# Generating iPSCs with a High-Efficient, Non-Invasive Method—An Improved Way to Cultivate Keratinocytes from Plucked Hair for Reprogramming

**DOI:** 10.3390/cells11121955

**Published:** 2022-06-17

**Authors:** Lisa S. Wüstner, Moritz Klingenstein, Karl G. Frey, Mohammad R. Nikbin, Alfio Milazzo, Alexander Kleger, Stefan Liebau, Stefanie Klingenstein

**Affiliations:** 1Institute of Neuroanatomy and Developmental Biology (INDB), Eberhard Karls Universität Tübingen, 72074 Tübingen, Germany; lisa-sophie.wuestner@student.uni-tuebingen.de (L.S.W.); moritz.klingenstein@uni-tuebingen.de (M.K.); georg.frey@student.uni.tuebingen.de (K.G.F.); mohammad-reza.nikbin@uni-tuebingen.de (M.R.N.); alfio.milazzo@uni-tuebingen.de (A.M.); stefan.liebau@uni-tuebingen.de (S.L.); 2Department of Internal Medicine I, University Medical Center Ulm, 89081 Ulm, Germany; alexander.kleger@uniklinik-ulm.de

**Keywords:** plucked human hair, hair follicle (HF), keratinocytes, induced pluripotent stem cells (iPSCs)

## Abstract

Various somatic cell types are suitable for induced pluripotency reprogramming, such as dermal fibroblasts, mesenchymal stem cells or hair keratinocytes. Harvesting primary epithelial keratinocytes from plucked human hair follicles (HFs) represents an easy and non-invasive alternative to a fibroblast culture from invasive skin biopsies. Nevertheless, to facilitate and simplify the process, which can be divided into three main steps (collecting, culturing and reprogramming), the whole procedure of generating hair keratinocytes has to be revised and upgraded continuously. In this study, we address advancements and approaches which improve the generation and handling of primary HF-derived keratinocytes tremendously, e.g., for iPSCs reprogramming. We not only evaluated different serum- and animal-origin-free media, but also supplements and coating solutions for an enhanced protocol. Here, we demonstrate the importance of speed and accuracy in the collecting step, as well as the choice of the right transportation medium. Our results lead to a more defined approach that further increases the reliability of downstream experiments and inter-laboratory reproducibility. These improvements will make it possible to obtain keratinocytes from plucked human hair for the generation of donor-specific iPSCs easier and more efficient than ever before, whilst preserving a non-invasive capability.

## 1. Introduction

Stem cells present two major properties: a capacity for proliferation and an ability to differentiate into various cell types [1,2]. The general potential of stem cells in the body can vary greatly. Most stem cell populations are multipotent and give rise to various cell types of a distinct organ, tissue or cell layer [3]. In contrast, pluripotent stem cells can theoretically proliferate indefinitely and differentiate into all cell types and tissues of the three germ layers [4]. With the development of human-induced pluripotent stem cells (iPSCs), a completely new method for stem cell research arose, diminishing ethical concerns since embryonic stem cells can be replaced by iPSCs [5]. IPSCs also hold great potential in terms of generating individual iPSCs. Furthermore, human iPSCs are a powerful tool for answering developmental questions, for drug screening, and for in vitro disease modelling or therapeutical approaches, generally with the ability to reduce or replace animal testing with, e.g., the goal to contribute to a better understanding and treatment of diseases [6,7].

The generation of iPSCs is generally based on an adult, somatic cell source being induced to an early, embryonic-like developmental state. Next to the widely used mesoderm-derived skin fibroblasts, several other primary cell sources have been described as suitable for reprogramming to iPSCs [8]. Each somatic cell source has its own advantages and disadvantages. Procedural invasiveness in obtaining primary cells, transportation, cultivation in vitro and reprogramming efficiency are challenges to take into account when choosing the appropriate somatic cell source [8].

More than a decade ago, the first protocol providing a method to generate iPSCs from keratinocytes obtained from plucked hair was published [9]. This publication presented a practical, non-invasive approach to obtain samples from voluntary donors and patients independent of age, color of skin, sex or religious conviction. Moreover, the harvesting of this somatic source is easy to perform by non-medical personnel and provides an effective way to generate iPSCs. Until then, like dermal fibroblasts, keratinocytes were usually extracted from the skin, which necessarily includes invasive skin biopsies [10,11]. Over the years, many modifications and improvements were achieved in the collection, transportation and cultivation of plucked human HFs and keratinocytes [12]. An improved and reliable protocol for isolating keratinocytes from patient hair samples facilitates the generation of patient-specific stem cells and further experimental approaches. Here, we want to share our new insights into the efficient cultivation of HF-derived keratinocytes for the consecutive successful generation of iPSCs. This ranges from optimal collecting and transport solutions to productive cultivation and reprogramming deliverables.

## 2. Materials and Methods

### 2.1. Plucking and Preparation of Human Hair Follicles (HFs)

This method has been previously described in detail [13]. In short: HFs were obtained from seven different, healthy donors (Appendix A). Hair was plucked from the occipital area of the scalp, and the HFs were transferred into the transportation medium immediately. Of great importance was the use of HFs with an intact and clearly visible outer root sheath (ORS) (for an example of an intact ORS, see Figure 1).

### 2.2. Cultivation of HF-Derived Keratinocytes

After cutting the hair 1–2 mm distal to the ORS, the HFs were put in pre-coated culture vessels with conditioned mouse embryonic fibroblast (MEF) medium (MEFCM) and incubated in a humidified atmosphere with 5% CO_2_, 5% O_2_, at 37 °C. Deviations from the standard protocol are explicitly mentioned in the results section. The standard protocol has been previously published [13]. A summary of the described methods: after the outgrowth of the first keratinocytes, the medium was changed from MEFCM to the EpiLife medium + HKGS (ThermoFisher, Waltham, MA, USA) supplement according to the manufacturer’s protocol. Deviations from the standard protocol are explicitly mentioned in the results section.

### 2.3. Media Used in This Study

Appendix A summarizes all media and supplements used for the keratinocyte culture in this study. Culture media were prepared according to the suppliers’ recommendations. During reprogramming, cells were cultured in hiPSCs medium: KnockOut-DMEM supplemented with 20% KnockOut SerumReplacement, 1% Antibiotic Antimycotic, 100 μM NEAA, 1% GlutaMAX, 100 µM β-Mercaptoethanol (all Thermo Fisher, Waltham, MA, USA), 10 ng/mL FGF2 (Cell Guidance Systems, Cambridge, UK), 10 µM Y-27632 (Selleckchem, Planegg, Germany) at 5% CO_2_, 5% O_2_, at 37 °C.

### 2.4. Coating Solution and Mounting Droplet Used in This Study

Appendix A summarizes the coating solutions used in this study. Coating solutions were prepared according to the suppliers’ recommendations. 700 µL of the coating solution has been used to cover the bottom of a T25 flask, which was then incubated for 1 h at 37 °C in a humified incubator. For the droplets, Matrigel Basement Membrane Matrix, LDVE-free (Corning, New York, NY, USA) was either used pure or diluted in EpiLife medium at the ratio 1:5, 1:2.5. Each HF was covered with 10 µL of the coating solution and incubated for another 2–3 h in the incubator. The coating matrix kit was not used together with any Matrigel droplets. After the second incubation step, 1 mL of appropriate culture medium was added and incubated for another 48 h until the next medium change.

### 2.5. Defining Outgrowing Keratinocytes from HF “Circle Condition”

For analyzing outgrowing keratinocytes from plucked HFs objectively, a defined circle of 0.3 mm diameter was used (Figure 2). This specific area was chosen as it marks the average length of the ORS. Due to previous observations, it marks a point where the HF will not self-detach easily anymore and the outgrowing cells will not stop proliferation. In addition to that, it is a representative measure for the speed of proliferation and the efficiency of the analyzed cultivation condition. Therefore, the circle was placed under the culture vessel to check whether the keratinocytes fully covered the area of the circle, which was defined as “circle condition achieved”. HFs were monitored individually for at least 14 days. Proliferating keratinocytes that failed to fill the area after two weeks were considered as “circle condition not achieved”. If no keratinocytes grew from a HF after one week, the experiment was excluded.

### 2.6. Reprogramming of Keratinocytes

Keratinocytes were enzymatically detached using TrypLE Express (Thermo Fisher, Waltham, USA) for 5 min at 37 °C. They were used for the standard protocol after reaching 70% confluency and stored in 800 µL SynTha-Freeze (Thermo Fisher, Waltham, MA, USA) in liquid nitrogen until further reprogramming. Cells were also directly used, termed “fresh cells”, for further reprogramming using two different approaches: the incubation of keratinocytes with viral particles for 48 h with or without an additional centrifugation step. The lentiviral transduction has been previously described in detail [13]. In short: lentiviral particles were obtained from transfected Lenti-X 293T cells for 4 h in a serum-free medium using 8 µg lentiviral pRRL.PPT.SF.hOKSMco.idTom.pre FRT plasmid [14] together with 5.5 µg psPAX2 and 2 µg pDM2.6 plasmid using polyethylenimine (Sigma Aldrich, St. Louis, MO, USA). The medium is changed to a serum-containing MEF medium and the supernatant can be collected 2 and 4 days after transfection. The concentration of viral particles was performed using a Lenti-X concentrator (Takara, Kusatsu, Japan) according to the manufacturers protocol. The viral particles can be stored at −80 °C. Differing from this method, the “Spinfection” technique uses a centrifugation step after the addition of the lentiviral particles. Instead of incubating cells for 2× 24 h with the viral particles, the keratinocytes were centrifuged for 45 min at 32 °C with 1500 rpm for 2 days in a row with the viral particles, which were removed after centrifugation by a complete exchange of media. 48 h post-transduction, the cells were detached and transferred onto a MEF feeder layer and incubated with hiPSC medium until iPSC colonies started to appear. Upon reaching the appropriate size, colonies were picked mechanically and transferred into a feeder-free culture system based on a Matrigel coating with Essential 8 medium (Thermo Fisher, Waltham, MA, USA).

### 2.7. Quality Control of Generated iPSCs

Newly generated iPSC lines have to undergo various quality checks to proof their full pluripotency. Besides the characteristic cell morphology (Appendix A), which is marked by round colonies with tightly packed cells with a prominent nucleus, the expression of specific gene and cell surface markers has to be checked. On the RNA and protein levels, pluripotency markers as well as markers defining the potential for the differentiation capacity into all three germ layers can be found in Appendix A. The integrity of the genome has to be validated using karyotyping, as well by addressing the epigenetic state using bisulfide sequencing, which has been tested on previous lines [15] and is performed routinely at our institute. The check for a mycoplasma-free cell culture is routinely performed in our lab four times a year using the PCR Mycoplasma Detection Kit (Biozol, Eching, Germany) according to the manufacturers protocol.

### 2.8. Histological H&E Staining

Cross- and longitudinal cryosections of HFs were stained with Haematoxylin and Eosin (H&E). Samples were first treated with filtered Haematoxylin for 4 min, followed by a washing step using running tap water for approximately 2 min. Subsequently, the slides were treated with filtered Eosin solution for 2 min and washed with sterile water. After the treatment with an alcohol gradient (70–95–100–95–70%, 30 s each), the slides were placed in Xylol for an additional 30 s. The slides were mounted with DePeX mounting medium (VWR, Radnor, PA, USA) and stored at room temperature.

### 2.9. Alkaline Phosphatase Staining

To assess reprogramming efficiency, the cells were stained for alkaline phosphatase three weeks after transferring the double infected keratinocytes onto the MEF feeder layer. A Sigmafast BCIP/NBT tablet (Sigma-Aldrich, St. Louis, MO, USA) was diluted in 10 mL of water and added to the cells with the exclusion of light. After approximately 25 min, the reaction was stopped, and the cells were stored in DPBS^−/−^ at 4 °C. The colonies per well were counted, and the mean was calculated.

### 2.10. Immunofluorescence

Cells were fixed with 4% PFA for 15 min at room temperature followed by two washing steps with DPBS^−/−^ for 5 min each. After blocking with skimmed milk blocking solution for 1 h, primary antibodies were added. Primary antibodies were diluted in a blocking buffer and incubated for 2 h at room temperature in a humidified chamber. Secondary antibodies were diluted 1:1000 together with DAPI 1:100 (Sigma Aldrich, St. Louis, MO, USA) in DPBS^−/−^ and incubated for an additional 1 h at room temperature under the exclusion of light. Sections were mounted with Mowiol (Carl Roth, Karlsruhe, Germany). Further information about the blocking solution and antibodies can be found in Appendix A.

### 2.11. Quantitative RT-PCR

For the analysis of the germ layer differentiation, total RNA was isolated from differentiated cells using RNeasy MiniKit (Qiagen). OneStep quantitative real-time PCR was performed on a StepOne Plus instrument using QuantiFast SYBR Green RT-PCR Kit (Qiagen) according to the manufacturer’s protocol. Relative gene expression of *OCT4* (Qiagen, #QT00186759), *NANOG* (Qiagen, #QT01025850) and *SOX2* (Qiagen, #QT00237601) was calculated to the expression of the housekeeping gene *GAPDH* (Qiagen, #QT00014462) (Appendix A) (Appendix C). 

### 2.12. Microscopic Imaging

Images of histological H&E stainings were obtained using a PrimoVert light microscope (Zeiss, Jena, Germany). For bright fields, an EVOS FL cell imaging system (Thermo Fisher, Waltham, MA, USA) was used. Immunofluorescence pictures were obtained and analyzed using the Axio Imager M2 microscope with the AxioVision software (Zeiss, Jena, Germany).

### 2.13. Statistics

Statistical analysis was performed using Excel (Microsoft, Redmond, WA, USA) and JMP (JMP Statistical Discovery, Cary, NC, USA). For comparison of the proportional numbers of outgrowing HFs, pairwise Chi2 tests were performed with a Bonferroni adjusted α = 0.005. For the statistical analysis of differences between the time needed for outgrowth, a Wilcoxon rank sum test was performed. For the investigation of the time needed to fulfil the circle condition, the duration from the individual timepoint of primary outgrowth was compared and analyzed by performing a Wilcoxon rank sum test. The final sample size (n) is indicated in the corresponding results by means of the total HFs used. For the calculation of the HF diameter, a TTEST with the following *p*-value was performed: *p* < 0.05 *, *p* < 0.01 **, *p* < 0.001. For most experiments the number of single repeated experiments was too low for statistical analysis.

## 3. Results

### 3.1. Extraction of Human Hair Follicles (HFs)

#### 3.1.1. Anatomical Structure of Human HFs Used for This Study

The hair organ consists of concentric layers and sheaths, with the ORS as the outermost, followed by the inner root sheath (IRS) and the keratinized hair shaft in the center (Figure 1A). HFs can be plucked from the scalp of nearly all humans. Keratinocytes are specialized cells which lie in distinct areas of the hair follicle and are necessary for the reproduction of lost HFs during the normal hair cycle [16]. Keratinocytes used for our approaches mainly derive from the outer root sheath (ORS) (Figure 1A,B).

#### 3.1.2. Dehydration of Plucked HFs

To obtain keratinocytes from human hair, the first obstacle is the quick and precise extraction of hair roots with an intact ORS. Plucking too slowly can lead to dehydration of the ORS and the keratinocytes within, which is one of the major causes of a failure to grow keratinocytes from the HF in a culture. For better visualization of the dehydration process, a video was recorded showing freshly plucked hair exposed to air for a maximum of 5 min (Video S1). Figure 1C shows an HF exposed to air for 120 s compared to a freshly plucked HF (Figure 1B). Macroscopically, the shriveled ORS and the drastically reduced diameter of the hair organ caused by dehydration can be easily observed.

To evaluate if the exposure time to ambient air has an influence on a successful outgrowth, different conditions were monitored. After exposure to ambient air for 10 s, 30 s, 60 s, 90 s and 120 s, the importance of “air time” becomes clearly visible (Figure 1D–I). With the naked eye, we could observe that HFs confronted with ambient air for 90 s (n = 2) or 120 s (n = 3) were totally dehydrated (100%). Moreover, 70% of the HFs with short exposure to ambient air for only 10 s (n = 3) already dried out dramatically (Figure 1D). This is a substantial percentage considering the real-life conditions during extraction. After the plucking of hair, it is highly recommended to check the suitability of the hair root. Checking and transferring the hair root to the transport vessel makes a 10 s handling time very realistic. Interestingly, when the hair root is added directly (0 s) to the transportation medium and exposed to the air afterwards, a different result is obtained (Figure 1D,F). When the HFs were exposed to ambient air after transportation for 45 min the in DMEM medium, less severe observations were made. By the naked eye, 50% of the tested HFs were dehydrated after exposure to air for 30 s (n = 4), 90 s (n = 4) and 120 s (n = 4). The exposure to air for 10 s (n = 3) seemed not to influence the diameter of the ORS (Figure 1E). None of the visibly dehydrated HFs were able to generate keratinocyte outgrowth within 14 days, independent of their exposure to ambient air occurring before or after the transportation of the sample in the DMEM medium.

#### 3.1.3. Outgrowth from HFs before and after Transportation

Next, we checked for the outgrowth behavior of HFs. For the first condition, HFs were exposed to air for 10 s to 120 s of exposure time, followed by 45 min of transportation in standard DMEM medium. Here, we could see that from HFs which were exposed to air for 90 s (n = 4) or 120 s (n = 3), no cells were able to grow out, which fits to the drying conditions observed earlier (100%). However, exposure to air does not seem to influence the outgrowth capacity of HFs when kept in air for 10 s to 60 s (Figure 1F). Procedures were then performed vice versa: HFs were first stored for 45 min in DMEM transportation medium, followed by exposure to air for from 10 s to 120 s. HFs exposed to ambient air for 90 s (n = 5) and 120 s (n = 4) fail, whereas HFs exposed to the other time points were not statistically significantly decreased in their ability to generate an outgrowth of keratinocytes (60 s, n = 3). In 50% (10 s, n = 4) or 75% (30 s, n = 4) of the analyzed HFs, and cells were able to grow out (Figure 1G).

#### 3.1.4. Diameter of the HFs before and after Transportation

For a more objective approach, diameter measurements of DIC images of hair roots after different periods of air exposure time were performed (Figure 1H). The diameter of freshly plucked HFs shrunk significantly after 30 s (n = 4, *p* < 0.05), 60 s (n = 4, *p* < 0.01), 90 s (n = 4, *p* < 0.01) and 120 s (n = 4, *p* < 0.001) when exposed to ambient air in comparison to unexposed HFs. Equivalently, the diameter of exposed HFs after 45 min of transportation time showed a significant reduction compared to the controls (Figure 1I). Here, the diameters of the HFs were significantly reduced after 30 s (n = 4, *p* < 0.05), 60 s (n = 4, *p* < 0.05), 90 s (n = 4, *p* < 0.05) and 120 s (n = 4, *p* < 0.05) when exposed to ambient air.

Main results for the extraction of human hair follicles:➔Dehydration of ORS can be observed with the naked eye.➔Plucked HF samples must be transferred to a transportation medium immediately without extended exposure to air.

### 3.2. Transport and Safekeeping of Human Hair Follicles

#### 3.2.1. Definition of the “Circle Condition”

After the successful extraction of HFs, choosing the right transport and safekeeping conditions are the next steps of great importance. Not only the right medium, but also the time interval between hair extraction and sample preparation in the laboratory is crucial and should be held as short as possible. The aim is to assess which of the possible available solutions and culture media are the best for the transportation of freshly plucked HFs. For analyzing the different transportation media, standard cell culture media (DMEM) are compared to other solutions which are available and commonly used in hospitals or medical practice. Thus, 0.9% NaCl, DPBS^−/−^ and a sterile water solution are the media to test. HFs are collected in those media and stored for different periods of time to monitor their viability, namely the outgrowth of keratinocytes, after storage. For this purpose, an objective evaluation tool for the successful outgrowth of keratinocytes from plucked hair is needed. Therefore, the so-called “circle condition” is used. In short, the coverage of the keratinocytes of a defined circular area in a specific time period is measured (for more details see Materials and Methods) (Figure 2).

#### 3.2.2. Transport Media

With this method we determined which of the tested media is the most suitable for the transport and long-term storage of HFs, especially when the viability of the plucked hair starts to decrease. With a maximum storage time of two hours at room temperature, DMEM (n = 7) shows the best results. With this cell culture medium, keratinocytes can be grown from all cultured HFs (100%), whereas only 3 of 7 HFs show growth after storage in DPBS^−/−^ (n = 7), and only 17% of the HFs stored NaCl (n = 6) fulfil the “circle condition”. No keratinocytes can be obtained from HFs stored in H_2_O (n = 14) (Figure 3A). The results are even more explicit after 24 h: neither DPBS^−/−^ (n = 12) nor NaCl (n = 12) or H_2_O (n = 6) deliver viable, growing HFs, although no visible changes occur to the ORS itself. In 100% of the HFs transported in DMEM (n = 12) for 24 h, keratinocytes are able to grow out and reach the “circle condition” (Figure 3B). As DPBS^−/−^, NaCl and H_2_O are not suitable for the transportation of plucked HFs, the following experiments were solely performed using DMEM as a transportation medium. Therefore, Figure 3C shows HFs stored in the DMEM medium for longer time periods. The number of outgrowing HFs tends to go down with a longer storage time, showing 90% outgrowth after 48 h (n = 10) and 50% after one week (n = 10). Storage of HFs in DMEM in the fridge at 4 °C did not achieve any growth (data not shown).

Main results for the transport and safekeeping of human hair follicles:➔DMEM is the only appropriate media for the transport and storage of plucked HFs when used at room temperature.➔To achieve the best results for outgrowing cells from HFs, the duration between the plucking and processing of the specimen should be a maximum of 2 h.➔It is still possible (50%) to cultivate keratinocytes from HFs even after one week in DMEM when stored at room temperature.

### 3.3. Appropriate Culture Medium

#### 3.3.1. Primary Culture Media in Comparison

Another major issue in culturing keratinocytes from plucked human hair roots is the choice of the appropriate culture medium. Here, we had to differentiate between two types of media: one for cultivating the plucked hair root until keratinocyte outgrowth, and a second for a keratinocyte culture after the initial appearance of migrating and dividing keratinocytes. DMEM may be the right medium for HF transportation, but the keratinocytes within the outer root sheath have special needs. Due to their high calcium sensitivity, the long-term cultivation of outgrowing keratinocytes in DMEM would lead to an arrest in their division as well as differentiation, yielding too few cells for reprogramming [17]. Different culture media are compared, focusing on the outgrowth ability and the subsequent cultivation of keratinocytes. In addition to the standard combination of media with and without serum, the use of stand-alone serum-free media is tested likewise (for comparison of all tested media, see Appendix A). For all experiments, plucked hair roots were directly transferred into DMEM prior to the HF preparation. The standard protocol currently used in our lab starts with the cultivation in mouse embryonic fibroblast (MEF)-conditioned media (MEFCM), followed by an EpiLife medium when the first keratinocytes grow out (Graphical Abstract). This protocol was first described in 2008 [9], and still, the EpiLife medium with an HKGS supplement is the most commonly used commercially available medium for keratinocyte propagation culturing. Due to the high demands of MEFCM production, we investigate whether a basic, non-conditioned MEF medium, which is the basic ingredient to produce MEFCM, is sufficient for reliable keratinocyte outgrowth. Accessorily, the EpiLife medium is tested to replace MEFCM, and two additional commercially available media, which are suitable for primary dermal keratinocyte cultivation, are added to the test group (Figure 4). KGM2 is a serum-free medium which contains bovine pituitary extract (BPE), similar to the EpiLife medium. BPE contains large quantities of growth factors and is often used in cell cultures to promote the growth of several types of human epithelial cells, since it is essential for mitotic activity [18]. DK-SFM is a defined, animal-origin-free medium. We tested this medium as it does not contain BPE, rendering the formulation invariable and robust and avoiding batch-to-batch variations [19].

Our results show that it is generally possible to cultivate keratinocytes in all five tested media. However, several differences can be observed in the relative number of outgrowing HFs, as well as the time needed until the first keratinocytes appear and their further proliferation is measured with the “circle condition”. With a storage time of two hours, 100% of the plated HFs are able to generate keratinocytes (n = 7) with MEFCM, and 91% of HFs grow out with the MEF medium (n = 22). In the EpiLife medium, in 84% of the HFs, expanding keratinocytes became visible (n = 25), whereas only 26% of the HFs in DK-SFM (n = 23) and 15% of the HFs in KGM2 (n = 20) showed a successful outgrowth of keratinocytes (Figure 4A). There was no significant difference in the number of outgrowing HFs between the first three culture media combinations (Chi2 test *p* > 0.005). However, significant differences (Chi2 test (1) = 16.3, *p* < 0.005) could be observed between the EpiLife medium and DK-SFM. Of note, there was no significant difference between DK-SFM and KGM2 (Chi2 test *p* > 0.005). After 24 h, 48 h and 7 d of storage in DMEM, no viable keratinocytes could be detected in either the EpiLife medium, DK-SFM or KGM2 (Figure 4B–D). In the analyzed timeframe, the MEFCM and MEF medium are able to provide an appropriate environment supporting the successful outgrowth of keratinocytes after longer storage periods. This experimental approach shows that DK-SFM and KGM2 are not appropriate media due to the low number of outgrowing HFs. In addition, the EpiLife medium was only convincing when it was used for HFs stored 2 h or less.

For the following experiments, plucked HFs were stored for a maximum of two hours in DMEM, and only the MEFCM, MEF medium and EpiLife medium were used.

#### 3.3.2. Application of the “Circle Condition” to the Compared Primary Culture Media

In the following experiments, we investigated the average period in days before keratinocyte outgrowth. In MEFCM, HFs showed a median outgrowth after 3.5 days (n = 20), 5 days in the MEF medium (n = 20) and 6 days in the EpiLife medium (n = 21) (Figure 4E). Next, we checked the culture conditions of the three selected media for the duration until the “circle condition” was fulfilled. Here, the three media showed no significant difference, with an average between 3.5 and 8 days until all keratinocytes covered the defined area of 7.07 mm^2^ to achieve the “circle condition”. Of note, after the initial outgrowth of keratinocytes from the ORS, the medium was changed to the EpiLife medium to compensate for the high calcium concentration in the MEFCM and MEF medium, strongly reducing the proliferation rate of keratinocytes [17].

#### 3.3.3. Secondary Culture Media in Comparison

Though KGM2 and DK-SFM were not considered to be suitable as primary media for the initial outgrowth of keratinocytes from HFs, they could still be efficient as secondary media after initial keratinocyte outgrowth. Testing this, we observed that with the DK-SFM medium, keratinocytes never achieved the full “circle condition” in our hands. Therefore, Figure 5 shows only KGM2 compared to the EpiLife medium. Cells in KGM2 (n = 6) required more time to achieve the “circle condition”, with an average of 11 days compared to the EpiLife medium’s (n = 8) average of 8 days (Figure 5A). Interestingly, the keratinocytes growing in KGM2 showed an altered morphological appearance. We observed an increased number of large, terminally differentiated keratinocytes with a flattened, polygonal morphology before reaching 70% confluency within the culture vessel (Figure 5C). In the EpiLife medium, dome-shaped, partly elongated, mitotically active and proliferative keratinocytes were mainly found (Figure 5B). Using immunofluorescence staining, we wanted to assess if an altered protein expression pattern could be identified between the smaller cells growing under the EpiLife medium conditions or the smaller and larger cells in KGM2 (Figure 5B–E). Keratinocytes were stained for cytokeratin (CK) 5, 10 and 14. Smaller cells, grown in the EpiLife medium and KGM2, were positive for CK14, CK5 and CK10, whereas larger cells from KGM2 were mostly negative for CK5. Ki67 staining also revealed that the larger cells, which were predominantly found in KGM2, were negative for the cell division marker Ki67 (Figure 5E). Of note, the smaller, mitotically active, and dividing keratinocytes growing in the EpiLife medium showed a positive Ki76 staining (Figure 5D).

Main results of appropriate culture medium:➔No significant difference in the number of successfully grown keratinocytes from HFs in the tested combinations of MEFCM + EpiLife medium, MEF medium + EpiLife medium or solely EpiLife medium.➔DK-SFM and KGM2 are not suitable for the generation and cultivation of keratinocytes from HFs.➔Most keratinocytes growing in KGM2 are large, polygonal-shaped and non-dividing cells.

### 3.4. Coating Conditions for the Successful Outgrowth of Keratinocytes

#### 3.4.1. Coating Conditions in Comparison

One major issue of cultivating HF-derived keratinocytes is the tendency of HFs to detach from the culture vessel before the outgrowth of keratinocytes due to shear stress while handling or media change. To successfully obtain keratinocytes, HFs must stay adherent to the coating until a certain number of cells have grown out. To overcome the problem of detaching and floating HFs, we investigated different coating solutions and mounting techniques of primary plucked HFs. In our current protocol, culture vessels are coated with Matrigel diluted 1:10 in the EpiLife medium, and plucked HFs are mounted within a drop of higher-concentrated Matrigel (1:5 dilution). Matrigel is a commonly used, complex protein mixture based on laminin and collagen IV [20]. For primary keratinocyte cultures, this matrix diluted in aa culture medium proved to be suitable, as it mimics the basement membrane connecting basal keratinocytes via hemidesmosomes in vivo [21]. Besides the animal origin and batch-to-batch variability, one major disadvantage is its undefined composition. This may lead, similarly to serum-based media, to variability in experimental results [20]. In addition to this established method, HFs were cultivated on a more defined, animal-origin-free collagen-I-based coating matrix kit, which is recommended for usage with a DK-SFM medium. Nevertheless, more HFs grew out when embedded in Matrigel compared to collagen I. The highest outgrowth rate, with 100% growing HFs, was achieved using MEFCDM (n = 7), followed by the MEF medium with 91% (n = 22) and the EpiLife medium with 84% (n = 25) on Matrigel. It seems that the usage of the collagen I coating matrix kit is less successful, as it achieved only 70% outgrowth with MEFCM (n = 10) and 77% with the MEF medium (n = 13), and only 14% of the HFs grew out in the EpiLife medium (n = 14) (Figure 6A). With the combination of the EpiLife medium and collagen I, most of the cells seem to stop proliferating.

#### 3.4.2. Morphological Alterations under Different Coating Conditions

However, not only does the number of outgrowing HFs differ, but the cell morphology is also different. Keratinocytes on the Matrigel coating showed the formation of a confluent cell layer around the outer root sheath of the HF (Figure 6B). Cells cultivated on the coating matrix kit, however, grew more isolated and scattered around the HF (Figure 6C). As our results show that Matrigel seems more suitable for the outgrowth of keratinocytes from HFs, independent of the used medium, we next checked if the Matrigel coating could be improved. Currently, 1:5 diluted Matrigel drops are used for mounting the HFs to prevent detachment and dehydration until the culture media is added. To achieve a reduced number of detached and floating HFs, a higher concentration of Matrigel with a dilution of 1:2.5 and pure Matrigel in comparison to cultivation without any droplets are tested. HF samples are monitored for detaching, the outgrowth rate and the morphology of the keratinocytes.

Main results of coating conditions for the successful outgrowth of keratinocytes:➔A 1:5 Matrigel concentration or no Matrigel droplets reduces the attachment of the HFs.➔ Higher concentrations of Matrigel increase the attachment fidelity of the HFs.➔Altered growth behavior and slowed outgrowth of keratinocytes can be observed with higher concentrated Matrigel droplets.

It was possible to generate primary keratinocytes from HFs mounted in all tested conditions with no significant differences (Figure 7). With 85%, most HFs grew out without additional drops (n = 13) or a Matrigel dilution of 1:2.5 (n = 6). Similar results were achieved with undiluted Matrigel 73% (n = 13), and a Matrigel dilution of 1:5 results in 71% of outgrowing HFs (n = 7) (Figure 7G). During the preparation and cultivation of HFs, it may happen that the HFs detach and float in the culture vessel. If the number of outgrown cells is high enough and if they are mitotically active, the detached HF poses no problem. A total of 14% of the HFs mounted in 1:5 Matrigel detached during the process of cultivation, either before or shortly after primary outgrowth (n = 7). 31% of the HFs detached when plated without additional drops (n = 13). In Matrigel (n = 11) and the Matrigel dilution 1:2.5 (n = 6), no detachment was observed (Figure 7H).

#### 3.4.3. Application of the “Circle Condition” to the Compared Matrigel Droplet Dilution

Of note, under these conditions, the time until outgrowth and the “circle condition” was prolonged (Figure 7I,J). It seemed that with increasing concentrations of Matrigel in the droplets, the time until outgrowth and the fulfilment of the “circle condition” was increasing likewise (Figure 7I,J).

However, not only were the outgrowth rate or the proliferation time affected by the Matrigel droplets, but even more obvious was the variable morphology and growth behavior of the keratinocytes (Figure 7A–F). The high-concentrated Matrigel droplets seemed to build net-like structures within the droplets. With the higher matrix stiffness and scaffold integrity of pure Matrigel, the keratinocytes tended to pile up and grow in a 3D-like manner or built net-like formations instead of forming 2D cell layers. However, once reaching the border of the Matrigel droplet, keratinocytes grew rapidly in the 2D layers.

### 3.5. Reprogramming Keratinocytes to iPSCs

After successfully generating HF-derived keratinocytes, we next compared the reprogramming efficiency with the EpiLife medium and KGM2 using two different methods for lentiviral transduction (Figure 8). Lentiviral reprogramming is a well-established technique to deliver the reprogramming factors to the host cell [22]. It has been shown that the lentiviral transfection of keratinocytes shows a very high reprogramming efficiency [13]. KGM2 is used for reprogramming since its media composition is comparable with the EpiLife medium.

Usually, when the keratinocytes reach approximately 70% confluency, the cells are frozen, stored and thawed for later reprogramming. In addition to this standard method, we tried the viral transduction of non-thawed, primary cells. Furthermore, the transduction of the cells using centrifugation (“spinfection”) with the viral particles is utilized in addition to the standard method without centrifugation. This centrifugation-based method has been described by the group that first described the successful generation of iPSCs from keratinocytes [23]. The viral particles are produced and stored in the EpiLife medium and KGM2.

In general, it is possible to cryopreserve keratinocytes for reprogramming, though after thawing the rate of cell death was higher. Figure 8D shows that KGM2 only produced iPSC colonies with spinfection and fresh keratinocytes. All other experiments using KGM2 showed no viable cells one day after seeding the keratinocytes. In all other conditions, the successful generation of iPSC colonies from viable keratinocytes (Appendix A) was possible. When counting alkaline phosphatase (AP)-positive stem cell colonies (Figure 8A–C) after reprogramming, first tendencies could be seen: the highest number of positive cells could be achieved with the combination of fresh, non-thawed cells infected in the EpiLife medium via spinfection (n = 1) (Figure 8D). The first iPSC colonies appeared nine days after seeding the infected cells on the MEF feeder layer. With the standard condition (cells frozen and thawed in the EpiLife medium with no centrifugation), considerably lower numbers of AP-positive colonies were observed (n = 2), and the colonies emerged later, around three weeks after transduction (Appendix B).

With an mRNA expression profile and immunofluorescence staining, the pluripotency of the HF-derived iPSCs was verified (Appendix A). All analyzed stem cells were positive for the nuclear pluripotency markers OCT4, SOX2 and NANOG on the protein and mRNA level, and showed positive staining for the surface pluripotency markers SSEA4, TRA1-60 and TRA1-81 (Appendix A) (Appendix C).

Main results of reprogramming keratinocytes to iPSCs:➔The most efficient way to generate keratinocyte-derived iPSCs with lentivirus induction is via fresh, non-thawed cells using the EpiLife medium and the spinfection protocol.➔All generated iPSC colonies are pluripotent and able to differentiate into all three germ layers.

## 4. Discussion

### 4.1. Why Use Keratinocytes as the Primary Cell Source?

Nowadays, all tissues provide somatic cells that can be used for reprogramming into human iPSCs. However, due to differences in their accessibility, invasiveness, efficiency, uncomplicated transportation and storage, and other features, some cell sources are more suited than others.

Keratinocytes as the primary somatic cell source shows a variety of advantages in comparison to other somatic cell types such as fibroblasts, urine cells or blood cells. Obtaining blood samples or fibroblasts from skin biopsies are invasive procedures which have to be performed under sterile conditions. Plucking hair is non-invasive, and sampling can be performed by non-medically trained personnel or even by the person themselves, independent of the location. The process of plucking hair does not require sterile conditions and equipment, which are indispensable for taking blood samples. Compared to other cell sources, including non-invasive cell sources such as epithelial cells from urine, another great advantage is the possibility of the immediate transportation of the specimen under room-temperature conditions, without time consuming pre-processing for several hours or days or specialized transportation conditions [24]. Still, the collection of HFs and the cultivation of keratinocytes have to be learned and the crucial steps have to be considered when working with keratinocytes as a somatic cell source. The experiments in this paper address these crucial steps to ensure an easy and reliable way to create iPSCs from keratinocytes.

### 4.2. Evaluating the Best Generation and Culture Conditions for Keratinocytes

The process of plucking, transferring into the storage solution and mounting HFs in the culture vessel is very time-consuming and can heavily influence the quality of the specimen. Our results show that just 10 s of air exposure almost exclusively results in dried HFs. Our data support the assumption that the most critical steps are the plucking and storage of HFs. Once dried, there is no evidence that this can be compensated by storage or rehydration in DMEM or other media. For the whole process of plucking hair roots, it is of great importance to work fast and transfer the specimen directly into the appropriate transportation medium. Of the different transportation media tested, which are commonly available in hospitals or medical practices, only DMEM at room temperature achieves the desired results. It is clearly visible that only DMEM provides the right requirements for the transportation and long-term storage of freshly plucked HF that leads to the successful outgrowth of keratinocytes. DPBS^−/−^ might only be usable for a short storage time in the laboratory and if no other transportation solutions are available, as only around 40% of the HFs are able to generate keratinocytes when stored for less than two hours. On the one hand, DPBS^−/−^ does provide an isotonic environment, but on the other hand, it lacks nutritional supplements to maintain long-term viability. For this purpose, DMEM is the best choice as it contains isotonic salt, amino acids, vitamins and other important components that provide an optimal environment. Even long-term storage is possible, since after seven days in DMEM at room temperature, keratinocytes can be generated. We recommend adding 1% antibiotic antimycotic solution to the transportation medium to prevent contamination of the cell culture, and the subsequent culture of the primary keratinocytes should be created without any addition of antibiotics, as this decreases the viability of the keratinocytes [25].

Serum-based media for the cell culture usually contain 10–15% fetal bovine serum (FBS), which is composed of several essential components for proliferation and cell maintenance, such as hormones, vitamins and growth factors [26]. Therefore, their benefits include faster growth and easier maintenance of the cultured cells. For keratinocytes, a serum-based medium can only be used for the primary outgrowth, as the rather high calcium concentration leads to terminal differentiation of the keratinocytes [23]. Besides the difference in calcium concentration, the increasing variability between lots and the unknown and undefined exact composition of the FBS and conditioned medium are negative side effects.

Therefore, we compared our serum-based protocol (MEFCM + EpiLife medium) with three defined media. KGM2 is serum-free and, similar to the EpiLife medium, contains BPE and only 0.06 mM CaCl_2_. In contrast, the animal-origin-free medium DK-SFM contains no BPE and has a concentration of <0.1 mM CaCl_2_. In our hands, the two alternatives KGM2 and DK-SFM do not provide an environment which is suitable for the outgrowth of keratinocytes from HFs. Especially for the cultivation of plucked HFs, the defined BPE-free DK-SFM showed no satisfying results. The MEF medium or MEFCM show similar results and are able to support keratinocytes in their outgrowth and further cell divisions without terminal differentiation. This indicates that the growth factors and nutrients provided by FBS seem to be sufficient for a successful generation of keratinocytes. Although the usage of the EpiLife medium prevents the high variability of FBS in cultures, this medium composition is still not completely defined and animal-origin free. The HKGS supplement contains a mixture of BPE, bovine insulin and bovine transferrin along with other hormones and growth factors. The EpiLife medium showed robust results for the outgrowth rate throughout all performed experiments, with the first appearance of cells and the shortest time to fulfil the “circle condition”.

We recommend the usage of this media depending on the specimen due to our expertise in the lab over many years. For standard keratinocytes from control persons, fast and reliable outgrowth can be achieved by the usage of the EpiLife medium. In addition, this also limits variations in the composition of the medium. Some individual specific HFs seem to be wispier and tend to have a less pronounced or shorter ORS. The timespan when the first keratinocytes appear is mostly prolonged, and they need more time to fulfil the circle condition. If working with rare specimens, the combination of the MEF medium for outgrowth and the EpiLife medium for proliferation seems to be the right choice. Here, we can achieve an outgrowth of the first cells after an average of five days. The time-consuming preparation of MEFCM does not seem to be necessary. The DK-SFM medium does not fulfil our requirements for the outgrowth and cultivation of keratinocytes, as only 26% of the HFs are growing out successfully. In addition, with this medium, it is not possible obtain a cell layer, filling the entire circle area. The generated keratinocytes are either too few or they are less proliferative compared to other conditions.

### 4.3. Which Coating Conditions Are the Best?

A major issue in cultivating HFs is the proper fixation to the coating matrix. Adherent HFs are crucial for the outgrowth of keratinocytes, and no cells can be obtained from initially detached and floating HFs. Our experiments, including aspects of the outgrowth rate, the morphological shape and the behavior of the keratinocytes, show the best results with a 1:5 Matrigel dilution. Here, the detachment rate of the HFs is lower than without droplets, and the morphology and outgrowth rate of the keratinocytes is better and higher than in less-diluted Matrigel (1:2.5 and pure). The combination of a coating matrix kit with the EpiLife medium is not recommended, as it showed the lowest outgrowth rate. While the HKGS supplement is used, the manufacturer recommends S7 supplementation in combination with the coating matrix kit. Preliminary experiments do not show a difference when using S7 instead of HKGS (data not shown). Here, further investigations have to be done, as the combination of the coating matrix kit together with the EpiLife medium and S7 would provide a defined environment for the cultivation of HF-derived keratinocytes.

### 4.4. How to Achieve a Successful Lentiviral-Based Reprogramming of Keratinocytes?

As one of the most efficient and well-established reprogramming strategies, we used lentiviral-based reprogramming to prove the reprogramming efficacy of the respective conditions. We used two infection methods (standard and “spinfection”) and two media (EpiLife medium and KGM2). In addition, we tested whether keratinocytes can be used after cryopreservation. Indeed, the reprogramming efficiency is much higher and the time until first stem cell colonies appeared is also shorter with the combination of fresh cells with the EpiLife medium and the spinfection method. The generated iPSCs via spinfection can be verified as pluripotent. It also seems that the EpiLife medium is more suitable for transfection than KGM2 in comparison. Overall, the EpiLife medium performs best in the cultivation of HF-derived keratinocytes. Therefore, this medium might be the one to use, especially considering more recent, integration-free reprogramming methods that are applicable to keratinocytes [8,13].

## Figures and Tables

**Figure 1 cells-11-01955-f001:**
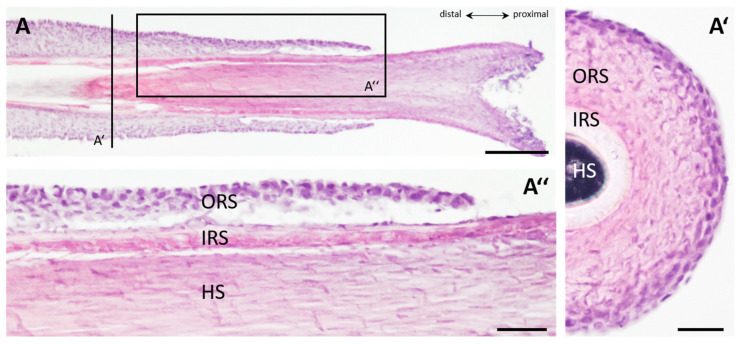
Anatomical structure and influence of ambient air exposure to plucked human hair follicles. (**A**) H&E staining of a sectioned, plucked human hair follicle. Longitudinal section with higher magnification of the proximal suprabulbar portion identifies the different layers shown in (**A″**) (black box). (**A′**), Distal cross-section of a human hair follicle (HF) displayed by a black line depicted in A. Outer root sheath (ORS) indicates the outermost layer with darker nuclear staining, followed by the inner root sheath (IRS) and the hair shaft (HS). (**B**) DIC picture of intact plucked human HF with clearly visible ORS and fully keratinized HS. (**C**) DIC picture of HF exposed to ambient air for 120 s with dried ORS. Subjective observation of visibly dried HFs before (**D**) and after transportation for 45 min in DMEM (**E**) exposed to air ranging from 10 s to 120 s. Percentage of outgrowing keratinocytes in relation to their exposure time to ambient air (10 s, 30 s, 60 s, 90 s, 120 s) before (**F**) and after (**G**) transfer to the transportation medium. Diameter of HFs exposed to ambient air for different time periods (10 s, 30 s, 60 s, 90 s, 120 s) relative to non-treated HFs before (**H**) and after (**I**) transferring into the transportation medium. TTEST *p* < 0.05 *, *p* < 0.01 **, *p* < 0.001 ***. N = number of single, independent experiments. Scale bar: 200 µm (**A**), 50 µm (**A′**,**A″**), 500 µm (**B**,**C**).

**Figure 2 cells-11-01955-f002:**
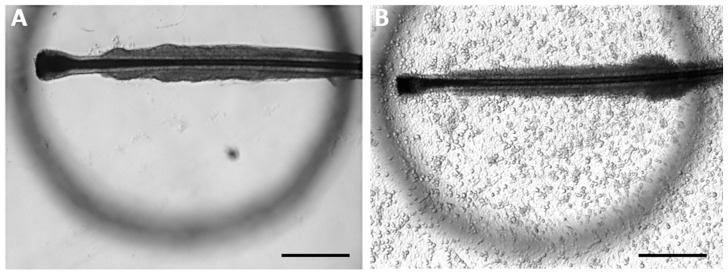
Light microscopy pictures of cultured HFs with the “circle condition”. A circle with a defined area of 0.3 mm in diameter is attached below the culture vessel. (**A**) Intact HF with the first outgrowing keratinocytes. Keratinocytes have not yet fulfilled the “circle condition”. (**B**) Keratinocyte cell layer reaching the size of the defined area of the circle and therefore categorized as achieved “circle condition”. This criterium is reached after an average of 6 to 8 days. Scale bar: 500 µm.

**Figure 3 cells-11-01955-f003:**
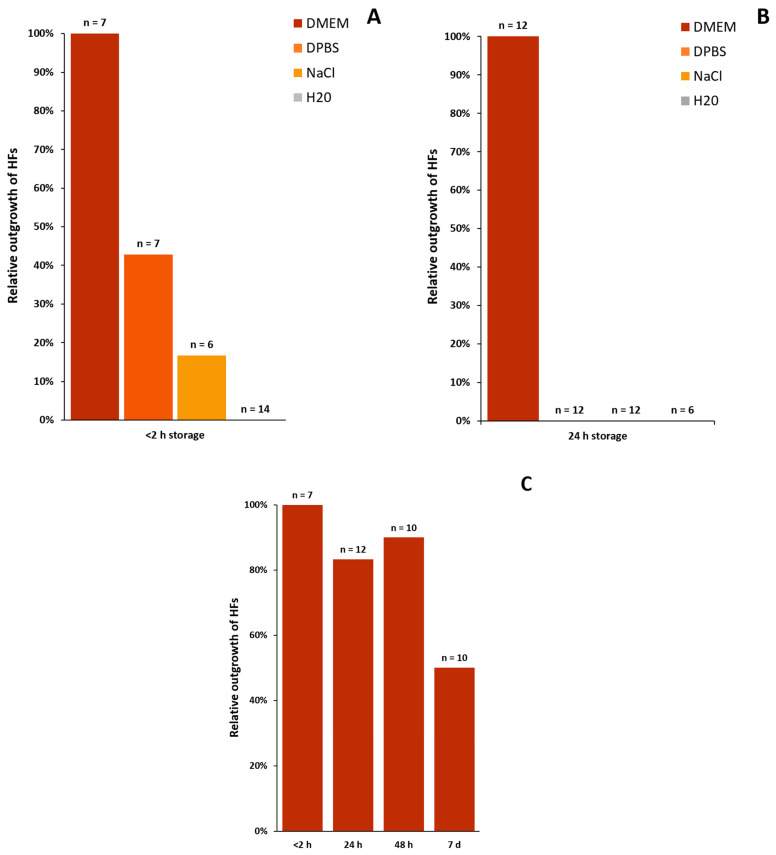
Outgrowth of HFs with respect to different transportation and storage conditions. (**A**) Relative outgrowth of keratinocytes from HFs after storage for a maximum of 2 h in either DMEM, DPBS^−/−^, NaCl or water (H_2_O). (**B**) Outgrowth after 24 h in the same four tested media. Keratinocytes stored in DMEM show the highest outgrowth rate. (**C**) Outgrowth of keratinocytes from HFs in percentanges after being stored in DMEM for a maximum of 2 h, 24 h, 48 h or 7 days (d). N = number of single, independent experiments.

**Figure 4 cells-11-01955-f004:**
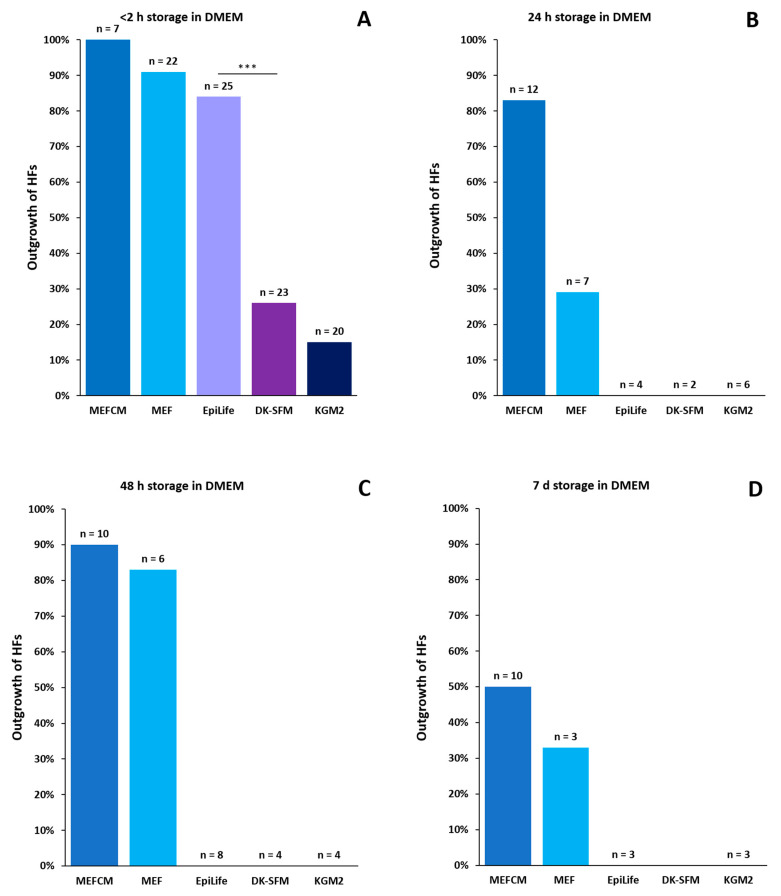
Influence of different culture media on keratinocyte outgrowth from HFs. Relative outgrowth of keratinocytes from HFs after cultivation in five different media (MEFCM, MEF medium, EpiLife medium, DK-SFM, KGM2) after <2 h (**A**), 24 h (**B**), 48 h (**C**) and 7 d (**D**) in DMEM transportation media. Both commercially available media DK-SFM and KGM2 showed the lowest outgrowth rate. Three different media (MEFCM, MEF medium, EpiLife medium) were compared for the time period until the first keratinocytes appear (**E**) and the time to fulfil the circle condition (**F**). Significance is indicated via asterisks (Chi2 test (1) = 16.3, *p* < 0.001). *p*-value: *p* < 0.001 ***. N = number of single, independent experiments.

**Figure 5 cells-11-01955-f005:**
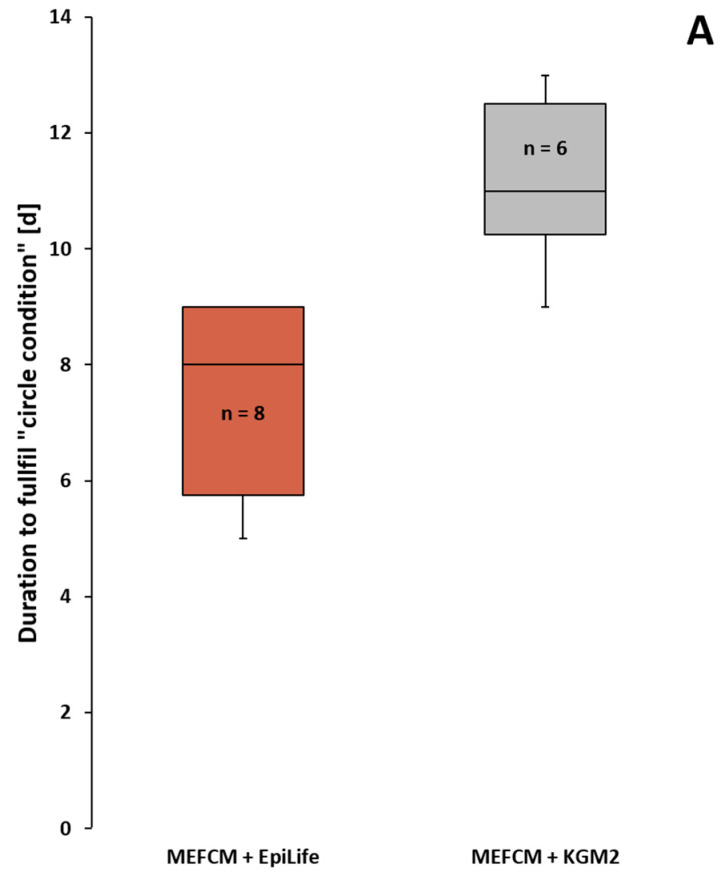
Comparison of KGM2 and EpiLife as secondary media. (**A**) Graph showing the days (d) until the keratinocytes achieve the “circle condition”. Keratinocytes were growing either in EpiLife or KGM2 media following the initial outgrowth in MEFCM. Overview DIC image of confluent keratinocytes cultured in either secondary EpiLife (**B**) or KGM2 (**C**) media. Notice the terminally differentiated keratinocytes in the KGM2 (**C**) with a large, polygonal-shaped cytoplasm. High magnification image of keratinocytes grown in MEFCM + EpiLife medium (**B′**–**B‴**) and MEFCM + KGM2 (**C′**–**C‴**) in a DIC image, immunostained for CK5,10,14 (green) and nuclear marker DAPI (blue). Ki67 staining (red) for the visualization of dividing keratinocytes either in Epilife medium (**D**) or KGM2 (**E**). Keratinocytes were counterstained for CK14 (green) and nuclear marker DAPI (blue). N = number of single, independent experiments. Scale bar: 400 µm (**B**,**C**), 50 µm (**B′**–**B‴**,**C′**–**C‴**,**D**,**E**).

**Figure 6 cells-11-01955-f006:**
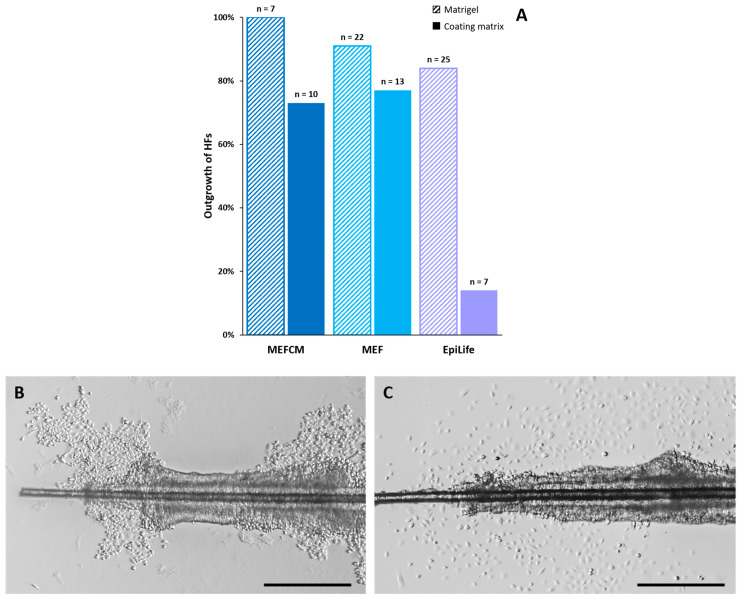
Growth behavior of keratinocytes on different coating solutions. (**A**) Relative primary outgrowth of HFs in different media (MEFCM, MEF medium, EpiLife medium) and coating conditions (Matrigel, collagen I coating matrix). (**B**) Representative DIC image of the distal part of an HF with confluent keratinocytes growing on a 1:10 diluted Matrigel coating after 9 days of cultivation. (**C**) Distal part of an HF with scattered keratinocytes outgrowing on the coating matrix kit after 9 days of cultivation. N = number of single, independent experiments. Scale bar: 500 µm.

**Figure 7 cells-11-01955-f007:**
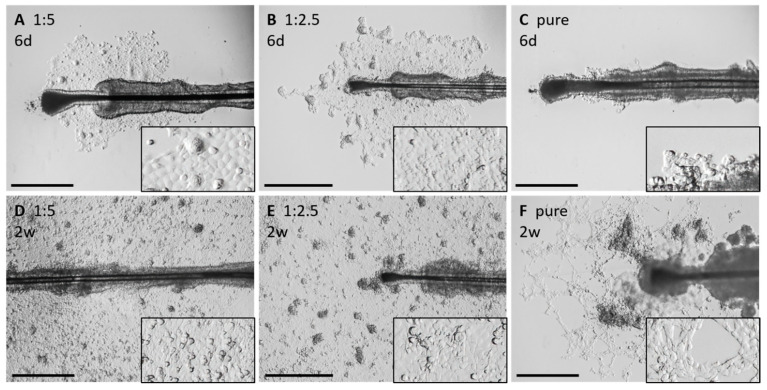
Influence of Matrigel droplet dilution on keratinocyte outgrowth. Keratinocyte growth behavior in different Matrigel mounting droplets (without drops (w/o); 1:5; 1:2,5; pure). DIC images of adherent keratinocytes growing under different coating conditions. After 6 days (6 d) of culturing (**A**–**C**) and after 2 weeks (2 w) of culturing (**D**–**F**). Relative outgrowth (**G**) and detaching (**H**) of HFs within four different Matrigel droplet dilutions. Analysis of the duration until the first outgrowth from HFs (**I**) and the time needed to fulfil the “circle condition” (**J**) with respect to the four different coating conditions. N = number of single, independent experiments. Scale bar: 500 µm.

**Figure 8 cells-11-01955-f008:**
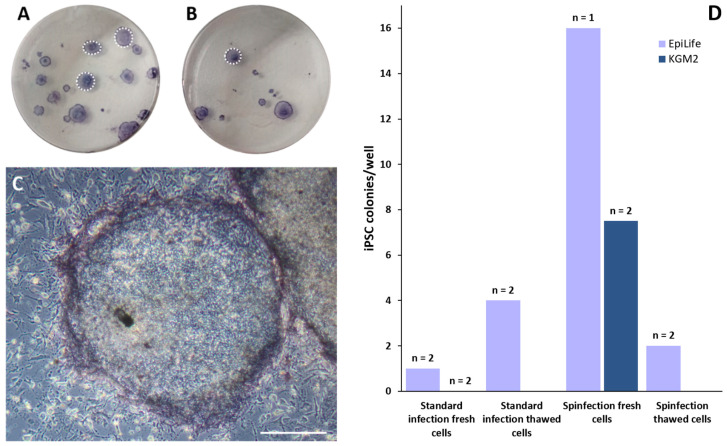
Reprogramming of HF-derived keratinocytes with different methods. Alkaline phosphatase (AP) staining of reprogrammed keratinocytes from the EpiLife medium (**A**) and KGM2 (**B**). Exemplary dotted lines mark the alkaline-phosphatase-positive iPSC colonies. (**C**) Higher magnification of AP-positive iPSCs growing on the MEF feeder layer. (**D**) Graph shows the amount of AP-positive iPSC colonies per well using different infection media and methods. N = number of single, independent experiments. Scale bar: 500 µm.

## Data Availability

Not applicable.

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
