# Peer review of "Generating iPSCs with a High-Efficient, Non-Invasive Method—An Improved Way to Cultivate Keratinocytes from Plucked Hair for Reprogramming"

_cells, 2022, doi:10.3390/cells11121955_

Round 1

Reviewer 1 Report

The authors aimed to re-fine the protocol for the reprogramming of keratinocytes obtained from plucked human hair. The authors state that this source of somatic cells can be used to replace fibroblasts being less invasive than starting from skin biopsies, today peripheral blood is used with great sucess. However, although the  procedures are well described, the authors do not discuss all the quality controls that most be performed on hiPSC before being used in down the line experiments. Moreover, lentiviral virus is used for the reprogramming of the keratnocytes, this approach requires extensive and more elaborate controls before being used. Have the authors tried other reprogramming alternatives? It would be nice to know how the authors plan to use the hiPSC neo-generated.

Author Response

Reviewer 1:

We thank the reviewer for the valuable suggestion and constructive critisism and we addressed all questions.

  1. The authors aimed to re-fine the protocol for the reprogramming of keratinocytes obtained from plucked human hair. The authors state that this source of somatic cells can be used to replace fibroblasts being less invasive than starting from skin biopsies, today peripheral blood is used with great success.

Answer: Using keratinocytes as the primary somatic cell source shows a variety of advantages in comparison to fibroblast, urine cells or blood cells. Obtaining blood samples, for example, is an invasive procedure which has to be performed under sterile conditions by medical trained personnel.

Plucking hair is non-invasive and sampling can be performed by non-medical trained personnel or even by the person oneself. We were able to pluck hair from new-born babies as well as patient and control subjects. Keratinocytes were successfully isolated and used for further experimental approaches. Another major big advantage is that the process of plucking hair does not require sterile conditions. For one specific project we plucked hair follicles from patients in the desert of Israel, impossible with most other cell sources. Another great advantage is the directly possible transport of the specimen without pre-processing for several days under room temperature conditions, for example advantageous when compared with urine. As we have several projects dealing with patient samples in different locations around the world, keratinocytes are most probably the most convenient primary cell source for the generation of iPSCs, at least from our own experience.

We added a new chapter in the discussion addressing the advantages and disadvantages of using keratinocytes as somatic cell source.

  1. However, although the procedures are well described, the authors do not discuss all the quality controls that must be performed on hiPSC before being used in down the line experiments.

We added a new chapter to the material and methods parts dealing with the controls we have performed with our newly generated hiSPCs. The associated data can be found in the supplementary Figures S1 and S2.

  1. Moreover, lentiviral virus is used for the reprogramming of the keratinocytes, this approach requires extensive and more elaborate controls before being used. Have the authors tried other reprogramming alternatives?

Some of the experiments for this specific project were conducted already several years ago, and we therefore used the lentiviral transduction for maintaining the coherence of this project. As the main focus of this publication is the transportation of hair follicles and the successful generation of keratinocytes, the reprogramming method is of secondary importance.

In our institute we have now completely switched to footprint-free Sendai reprogramming techniques. Currently, we are also establishing the reprogramming of keratinocytes using RNA approaches. Both methods work highly efficient for hair follicles, comparable to other cell sources

  1. It would be nice to know how the authors plan to use the hiPSC neo-generated.

In a variety of projects, neo-generated hiPSC lines are of utmost importance. Although, the CRISPRCas technique allows for the efficient generation of specifically engineered lines, the genetic background of disease models not only deal with single somatic mutations, but also with unknown modifiers and compensators. For that, person-specific hiPSC lines must be generated regularly. Other projects require a large number of different genetic backgrounds, sometimes even with specific needs. Additionally, the requirements for hiPSC lines change, for example as mentioned for the somehow obsolete lentiviral reprogramming, and we therefore have to generate new lines under new circumstances regularly.

Reviewer 2 Report

This is an interesting study that presents an improved approach for isolating and culturing keratinocytes from plucked human hair. The isolated keratinocyte can also be programmed to generate iPSCs. The work may offer foundation to further generate keratinocyte-derived iPSCs for skin or hair regeneration. 

  1. The presentation of figures and tables needs to be further improved. 
  2.  There is a room to improve the English writing. 
  3. The presentation of results also needs to be improved.

Author Response

Reviewer 2:

We thank the reviewer for this kind revision and improved the manuscript according to the reviewer`s suggestions. 

This is an interesting study that presents an improved approach for isolating and culturing keratinocytes from plucked human hair. The isolated keratinocyte can also be programmed to generate iPSCs. The work may offer foundation to further generate keratinocyte-derived iPSCs for skin or hair regeneration. 

  1. The presentation of figures and tables needs to be further improved. 

We have improved the quality of the tables and of all figures, especially figure 1, 2, 5, 6, 7, 8 and all supplementary figures.

  1.  There is a room to improve the English writing. 

The manuscript is revised regarding the English language throughout the text.

  1. The presentation of results also needs to be improved.

We have improved the presentation of the results. The summary of the main results of each chapter is highlighted and now fits better to the main text.

Reviewer 3 Report

The paper by L.S. Wüstner et al., “Generating iPSCs with a high-efficient, non-invasive method  The best way to cultivate keratinocytes from plucked human  hair for reprogramming “ is very simple and technical report but it might be important for those who work in this area. As referred in a paper many things are already done in this area but still some improvements are useful.

I have several comments for Authors: Page 88: 10 µM Y-27632 (Selleckchem) - Rock inhibitor listed as component of the culture media. Rocki can and should be used only for splitting hiPSCs but not as a constant component of the culture media. 

Page 93: Type of the Matrigel mast be stated -   Matrigel exist not of the animal origin but human grate 

Page 112 Keratinocytes were enzymatically detached - If Authors write a protocol can they specified  what enzyme and it concentration was  used in the study

Page 116 viral particles?? Author performed the lentiviral transduction.  Even they published a paper before ( Ref 13)  on reprogramming keranocytes -this information is important and need to be presented in detail.

Page  209-  tried out (100 %). Spelling - dried out 

Page 397 cell dividing marker Ki67. Spelling - ki67- Cell division marker 

FigS1 - of a very poor quality 

Author Response

Reviewer 3:

We thank the reviewer for the kind review and helpful suggestions. We addressed all issues raised in this review report and changed the manuscript accordingly.

The paper by L.S. Wüstner et al., “Generating iPSCs with a high-efficient, non-invasive method  The best way to cultivate keratinocytes from plucked human  hair for reprogramming “ is very simple and technical report but it might be important for those who work in this area. As referred in a paper many things are already done in this area but still some improvements are useful.

  1. I have several comments for Authors: Page 88: 10 µM Y-27632 (Selleckchem) - Rock inhibitor listed as component of the culture media. Rocki can and should be used only for splitting hiPSCs but not as a constant component of the culture media. 

We use Rock inhibitor for the reprogramming process in the hiPSCs medium. As soon the colonies reach a certain size and can be transferred to a feeder free system, we use Matrigel in combination with E8 medium for long term culture. That is totally correct, we, too, use the Rock Inhibitor only for spitting iPSCs. This information is added to the material and methods part.

  1. Page 93: Type of the Matrigel must be stated -   Matrigel exist not of the animal origin but human grate 

The Matrigel we use for this study is the “Matrigel Basement Mebrane Matrix, LDEV-free”. We listed it in supplementary table S4, we also added the information to the material and methods part.

  1. Page 112 Keratinocytes were enzymatically detached - If Authors write a protocol can they specified what enzyme and it concentration was used in the study

We use TrypLE Express in our protocols and added the enzyme and the protocol to the material and methods part.

  1. Page 116 viral particles?? Author performed the lentiviral transduction.  Even they published a paper before (Ref 13) on reprogramming keratinocytes -this information is important and need to be presented in detail.

A detailed protocol for the generation of lentiviral particles is added to the material and methods part.

  1. Page  209-  tried out (100 %). Spelling - dried out 

Spelling is corrected.

  1. Page 397 cell dividing marker Ki67. Spelling - ki67- Cell division marker 

Spelling is corrected.

  1. FigS1 - of a very poor quality 

We have improved the quality of Figure S1 and hope that it is now appropriate.
